CERN-TH-2024-177
CALT-TH-2024-038

# From data to the analytic S-matrix:
# A Bootstrap fit of the pion scattering amplitude

Andrea Guerrieri[a,b,c,d], Kelian Häring[a,e,f], and Ning Su[g,h]

[a] *CERN, Theoretical Physics Department, CH-1211 Geneva 23, Switzerland*
[b] *Department of Mathematics, City, University of London, Northampton Square, EC1V 0HB, London, UK*
[c] *Perimeter Institute for Theoretical Physics, 31 Caroline St N Waterloo, Ontario N2L 2Y5, Canada*
[d] *Dipartimento di Fisica e Astronomia, Universita degli Studi di Padova, Italy*
[e] *Institute for Theoretical Physics, University of Amsterdam, 1090 GL Amsterdam, The Netherlands*
[f] *Fields and Strings Laboratory, Institute of Physics,*
*École Polytechnique Fédéral de Lausanne (EPFL), CH-1015 Lausanne, Switzerland*
[g] *Walter Burke Institute for Theoretical Physics, Caltech, Pasadena, California 91125, USA*
[h] *Department of Physics, Massachusetts Institute of Technology, Cambridge, MA 02139, USA*

Quantum Chromodynamics (QCD) governs the strong interactions of hadrons, but extracting its physical spectrum remains a significant challenge due to its non-perturbative nature. In this Letter, we introduce a novel data-driven approach that systematically enforces the fundamental principles of *analyticity, crossing symmetry, and unitarity* while fitting experimental data. Our *Bootstrap Fit* method combines *S-matrix Bootstrap techniques* with *non-convex numerical optimization*, allowing for the construction of a scattering amplitude that adheres to first-principles constraints. We apply this framework to pion-pion scattering, demonstrating that it accurately reproduces low-energy predictions from Chiral Perturbation Theory ($\chi$PT) while also providing a non-perturbative determination of the total cross-section that is consistent with experiment.

A key feature of our approach is its ability to dynamically generate physical states, yielding a spectrum of resonances consistent with QCD. Most notably, we predict the existence of a *genuine doubly charged tetraquark resonance* around *2 GeV*, which could be observed in B-meson decays at LHCb. These results establish a robust new pathway for extracting hadronic properties directly from scattering data while enforcing fundamental physical constraints.

## I. INTRODUCTION

**S**CATTERING amplitudes are fundamental objects in quantum field theory. They contain information about the interaction among elementary particles and connect theoretical predictions with experimental observations. In strongly coupled theories such as Quantum Chromodynamics (QCD), the analytic structure of scattering amplitudes reveals crucial insights into hadronic states and resonances. However, extracting precise physical information from scattering data remains a formidable challenge due to the inherently strongly coupled nature of QCD at low energies.

The non-perturbative S-matrix Bootstrap [1, 2] provides a powerful, first-principles framework for studying hadronic amplitudes in a model-independent way, enforcing key physical constraints such as analyticity, crossing symmetry, and unitarity. It has been widely used to explore the spaces of scattering amplitudes, but it has not yet been applied to directly fit experimental data. In this work, we take a step in this direction by introducing the Bootstrap Fit, a method that integrates Bootstrap methods with non-convex optimization to extract hadronic properties while maintaining theoretical consistency.

Unlike traditional approaches such as Roy equations [3, 4], which generally rely on fixed $t$ dispersion relations and can only be used in a limited low-energy domain, our method offers a fully non-perturbative way to extract scattering amplitudes at all energies. By enforcing analyticity, crossing symmetry, and unitarity while fitting experimental data, the Bootstrap Fit provides a new pathway for modeling strong interactions without relying on perturbation theory and exploring its properties across all energies.

As a proof of concept, we perform this fit on pion-pion ($\pi\pi$) scattering, one of the most fundamental processes in QCD. Pions, as pseudo-Goldstone bosons of chiral symmetry breaking, provide an ideal testing ground for non-perturbative techniques. Using this novel approach, we achieve the following key results:

(i) Accurate low-energy predictions, consistent with Chiral Perturbation Theory ($\chi$PT).
(ii) Non-perturbative extraction of the total cross-section, in agreement with experimental data.
(iii) Self-consistent generation of QCD resonances, including the well-known states that couple to pions.
(iv) A concrete, experimentally testable prediction: a genuine doubly charged tetraquark resonance around 2 GeV, potentially observable in B-meson decays at LHCb.

Our method offers a systematic, data-driven strategy for modeling QCD amplitudes while rigorously enforc-

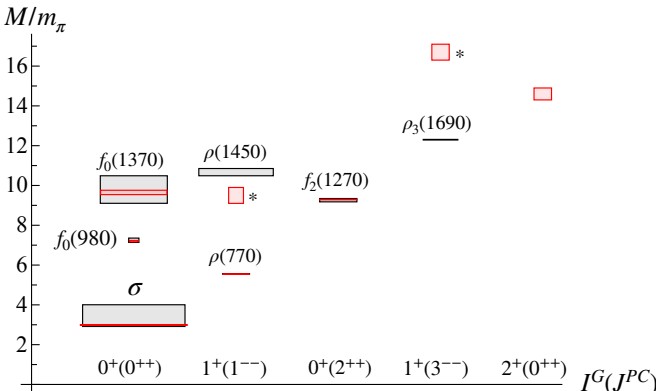

FIG. 1: The vertical extent of each box represents the uncertainty in the mass determination, while the horizontal width corresponds to $\Gamma/2$ with $\Gamma$ being the particle's decay width (uncertainties in the width are not shown). Mass and width are expressed in the same units. The gray boxes indicate the experimental spectrum, whereas the red boxes correspond to our Bootstrap estimate. Particles marked with an asterisk are still subject to numerical systematics, as discussed in Section V.

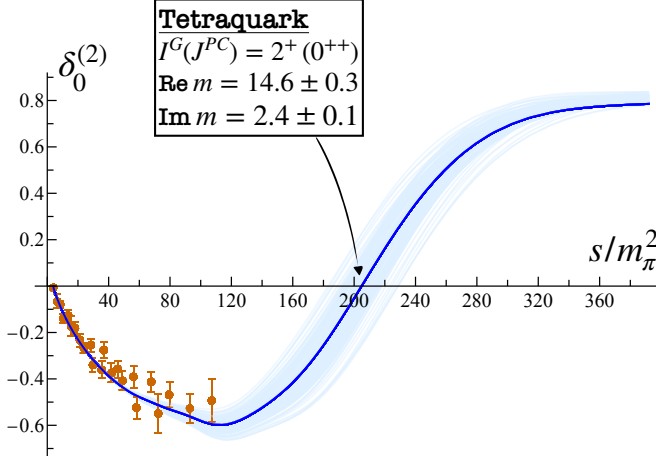

FIG. 2: High-energy behavior of the $S_2$ partial wave phase shift. The blue curve represents our best fit, while the light blue region includes amplitudes with suboptimal $\chi^2$ values (see Section IV). The phase shift exhibits a distinctive jump around $s \approx 200$, indicating the presence of a resonance. In the inset, we extract the mass parameter as $m = \sqrt{s^*}$, where where $s^*$ is the location of the zero in the complex $s$-plane of $S_0^{(2)}(s)$. All quantities are expressed in units of $m_\pi = 1$.

ing fundamental constraints. These results pave the way for future applications of Bootstrap methods in other hadronic scattering processes.

Our starting point is to consider the scattering amplitude of gapped pions. For simplicity, we choose the units by setting $m_\pi = 1$, neglecting isospin-breaking effects. In this setup, pions belong to the vector representation of $O(3)$, and the $2 \to 2$ scattering amplitude $\pi^a \pi^b \to \pi^c \pi^d$ takes the form:

$$\mathcal{T}_{ab}^{cd}(s,t,u) = \mathcal{A}(s|t,u)\delta_{ab}\delta^{cd} + \mathcal{A}(t|s,u)\delta_a^c\delta_b^d + \mathcal{A}(u|s,t)\delta_a^d\delta_b^c \tag{1}$$

where $s, t, u$ are the Mandelstam variables and $s+t+u=4$. By crossing symmetry, we also have $\mathcal{A}(s|t,u) = \mathcal{A}(s|u,t)$.

Pions are the lightest states of the QCD spectrum. A two-particle $\pi\pi$ state has $G$-parity $G = +1$, and parity $P = (-1)^J$, where $J$ is the spin of the particle. Only resonances with the same quantum numbers can be produced in a scattering experiment. In figure 1, the experimentally observed spectrum below 1.4 GeV is represented with gray boxes, adding only the $\rho_3$ above this energy.[1] From the amplitude $\mathcal{T}_{ab}^{cd}(s,t,u)$ constructed following the Bootstrap Fit procedure, we predict the spectrum and the low-energy parameters. In figure 1, we present in red the Bootstrap estimate of the spectrum. Both the mass and the width of the resonances match the experimental measurements, except for those denoted by an asterisk still affected by numerical systematics. The last column has the exotic quantum numbers $2^+(0^{++})$. There is no

such state in the PDG [5]. With its $I = 2$ quantum number, this state is a genuine Tetraquark [6, 7], predicted at $\sim 2$ GeV with a width of 600 MeV. As a prelude to our results, in figure 2 we show the predicted phase shift in the $S_2$ wave. At first glance, it would be difficult to anticipate from the experimental data that the phase shift would reverse direction at higher energies. This behavior is a genuine prediction of our Bootstrap procedure which incorporates full crossing and unitarity of the scattering amplitude. Before discussing in detail the physics of this amplitude we shall describe the steps of our strategy.

The rest of this paper is structured as follows. In Section II, we discuss the S-matrix Bootstrap ansatz, where we solve a semi-definite optimization problem to construct a candidate amplitude $\mathcal{A}_\Theta^{\text{ansatz}}(s|t,u)$ depending on a small set of free parameters $\Theta$. Section III describes our Particle Swarm Optimization (PSO) algorithm, which minimizes the $\chi^2$ function for the parameters $\Theta$ by comparing it with experimental and lattice data. In Section IV, we present the results of our Bootstrap Fit. We then test our amplitude in Section V against Chiral Perturbation Theory ($\chi$PT) and other experimental data, finding strong agreement. Finally, in Section VI, we discuss potential improvements and future research directions.

## II. BUILDING THE FIT MODEL

A key element of our methods is the numerical non-perturbative S-matrix bootstrap. This method was introduced in [1, 2] as a way to study the space of amplitude, deriving bounds on low-energy observables. It has since been applied in various physical systems [8–25]. Specifi-

---

[1] To express the spectrum in dimensionless units we use $m_\pi = 137.3$ MeV, the average between charged and neutral pions.

cally, its application to the scattering amplitudes of pions was revived in [26] and extended to the massless case in [27]. In [28], the authors refined the allowed region found in [26] using additional positivity constraints at low energy. By performing hypothesis testing using relative entropy, they selected the region of low energy parameters that matches QCD low energy data and $\chi$PT. In [29], they extracted the spectrum finding hints of emerging Regge trajectories. In [30, 31], the authors used several low energy constraints below the two-particle threshold, and form factors constraints computed in perturbative QCD at energy above 2 GeV, to select a region in parameter space that nicely agrees with experimental phase shifts and low energy data. Simultaneously, the Bootstrap of large-$N$ $\pi\pi$ amplitudes was kicked off in [32] and [33], and generalized by including photons and matching with the chiral anomaly in [34]. Finally, including a minimal amount of spectrum assumptions, a Bootstrap candidate for the large-$N$ QCD amplitude was found in [35].

In the following, we layout the S-matrix Bootstrap construction of the non-perturbative $\pi\pi$ scattering amplitude used in this Letter.

### A. The analytic Bootstrap ansatz

We parametrize the pion amplitude with the $\rho$-ansatz [2, 26] constructed to be analytic and crossing symmetric.[2] For our problem, it is convenient to introduce an extended version of the $\rho$-ansatz, the underline{multi-foliation} ansatz. We call underline{foliation}, a sum of the form

$$\mathcal{F}_\sigma^N(s|t,u) = \sum_{0 \leq n+m \leq N} \alpha_{n,m}^{(\sigma)} \rho_\sigma(s)^n (\rho_\sigma(t)^m + \rho_\sigma(u)^m) +$$
$$\sum_{\substack{n+m \leq N \\ 1 \leq n \leq m}} \beta_{n,m}^{(\sigma)}(\rho_\sigma(t)^n \rho_\sigma(u)^m + \rho_\sigma(u)^n \rho_\sigma(t)^m), \quad (2)$$

where $\rho_\sigma(s) = \frac{\sqrt{\sigma-4}-\sqrt{4-s}}{\sqrt{\sigma-4}+\sqrt{4-s}}$ is a conformal map from the cut plane to the unit disk with center $\rho_\sigma(8-\sigma) = 0$. $\mathcal{F}_\sigma^N(s|t,u)$ is manifestly symmetric in $t,u$ which automatically enforces crossing symmetry. The multi-foliation ansatz is then obtained by summing over different centers $\sigma \in \Sigma$

$$\mathcal{A}^{\text{ansatz}}(s|t,u) = \sum_{\sigma \in \Sigma} \mathcal{F}_\sigma^{N_\sigma}(s|t,u). \quad (3)$$

The intuition behind this operation is simple. A single foliation $\mathcal{F}_\sigma^N$ approximates best the amplitude in the region $|s| \approx \sigma$. For single-scale problems, tuning $\sigma$ to

the desired scale is enough to achieve fast convergence [18, 22].[3]

The pion amplitude is a multi-scale function: it features chiral physics at the scale $s \approx 1$, several sharp resonances of different spin at $s \propto \Lambda_{QCD}^2$ between $s \approx 30$ and $s \approx 100$, and inelastic effects kicking in at the $K\bar{K}$ threshold $s \approx 50$.[4] Therefore, we choose $\Sigma = \{20/3, 30, 50, 86\}$. We detail the numerical implementation in Appendix A.

### B. The unitarity constraints

The ansatz (3) is not manifestly unitary for arbitrary values of the free parameters $\alpha_{n,m}^{(\sigma)}$ and $\beta_{n,m}^{(\sigma)}$. We impose unitarity as a numerical constraint on those coefficients. This is conveniently performed by projecting the amplitude on a set of partial waves diagonalizing the amplitude in angular momentum and flavor. First, we decompose the amplitude into the isospin channels

$$\mathcal{T}^{(0)}(s,t,u) = 3\mathcal{A}(s|t,u) + \mathcal{A}(t|s,u) + \mathcal{A}(u|s,t), \quad (4)$$
$$\mathcal{T}^{(1)}(s,t,u) = \mathcal{A}(t|s,u) - \mathcal{A}(u|s,t), \quad (5)$$
$$\mathcal{T}^{(2)}(s,t,u) = \mathcal{A}(t|s,u) + \mathcal{A}(u|s,t), \quad (6)$$

and then project into partial waves

$$t_\ell^{(I)}(s) = \frac{1}{32\pi} \int_{-1}^1 d\cos\theta \, P_\ell(\cos\theta) \mathcal{T}^{(I)}(s,\theta), \quad (7)$$

where $\cos\theta = 1 + 2t/(s-4)$. Finally, unitarity for partial waves becomes the probability conservation condition $|S_\ell^{(I)}| \leq 1$ for any $s > 4$, any $\ell$, and $I$, with $S_\ell^{(I)} = 1 + i\sqrt{\frac{s-4}{s}} t_\ell^{(I)}$.

Experimental data show a pronounced inelasticity around $s \approx 50$ at the $K\bar{K}$ threshold where the process $\pi\pi \to K\bar{K}$ goes on-shell, especially in the $S_0$ channel. The probability conservation must be then replaced by the stronger condition $|S_\ell^{(I)}| \leq \eta_\ell^{(I)}$. We impose this condition as an SDP inequality of the form[5]

$$\mathcal{U}_\ell^{(I)} = \begin{pmatrix} \eta_\ell(s) + \text{Im}\, S(s)_\ell & \text{Re}\, S(s)_\ell \\ \text{Re}\, S(s)_\ell & \eta_\ell(s) - \text{Im}\, S(s)_\ell \end{pmatrix} \succeq 0. \quad (8)$$

As for the functions $\eta_\ell^{(I)}(s)$, we use the mean value of the phenomenological parametrization for the "big-dip"

---

[2] Here we assume maximal analyticity. See [21, 36, 37] for a complementary Bootstrap approach that uses only the rigorous analyticity domain proven by Martin [38].

[3] By fast convergence we mean that there is a $\sigma$ which minimizes the difference between the true amplitude and the foliation $|\mathcal{A} - \mathcal{F}_\sigma^N|$ at fixed $N$.

[4] It is also possible to consider foliations with different branch points. This might help to encapsulate the $K\bar{K}$ threshold, and obtain a better fit of the data (see also section IV).

[5] The effect of inelastic constraints in the S-matrix Bootstrap has been discussed in [39]. The Bootstrap of full multi-particle processes has been only recently developed for branon scattering amplitudes in two space-time dimensions [40]. Inelasticities were also obtained as an output of a Bootstrap procedure in [41].

scenario discussed in [42].[6] These functions should be regarded as an additional phenomenological input.[7]

### C.  Soft theorems and Spectrum Assumptions

In [26], it was shown that the low energy parameters of the pion amplitude, such as scattering lengths, are well inside the allowed region determined by the general S-matrix Bootstrap constraints. To restrict the allowed region, it is necessary to impose additional conditions on the amplitude. To this end, we consider two types of constraints: soft theorems, and spectrum assumptions.

Soft theorems are the consequence of spontaneously broken chiral symmetry. If pions were massless, the amplitude would vanish $\mathcal{A} \to 0$ when any of the momenta of the particles become soft. As the pions are massive, this behavior is corrected by quantum effects and is no longer exact [44]. Nevertheless, the existence of low energy zeroes in the partial waves is a prediction of $\chi$PT [45]. For this reason, we impose that $t_0^{(0)}(z_0) = 0$, and $t_0^{(2)}(z_2) = 0$ for some $0 \leq z_0, z_2 \leq 4$. We refer to these constraints as chiral zeroes conditions.[8]

The physical spectrum is encoded into the position of pole singularities in the second sheet of the $2 \to 2$ scattering amplitude. Using the elastic unitarity condition, it is possible to relate those resonance poles to zeros of the $S$-matrix (not the amplitude!) in the first sheet.

To make an assumption on the spectrum, we impose conditions of the form $S_\ell^{(I)}(s_R) = 0$, which we call resonance zeros, where $s_R$ is the complex mass squared of the particle and $(\ell, I)$ its quantum numbers.

In this work, we assume the existence of four resonances: one for the $\rho(770)$ in the $P$ wave, the two $f_0$'s in the $S_0$,[9] and one $f_2$ in the $S_2$ wave. We only assume a minimal spectrum. All other resonances are not put in by hand but emerge dynamically from the fit, appearing as zeros of the $S$–matrix in the partial waves.

Finally, we stress that we remain agnostic about the numerical values of both chiral and resonance zeros. Their positions are predictions of the fit.

### D.  The target functional

The final step in constructing our fit model is selecting the Bootstrap target functional. While there are,

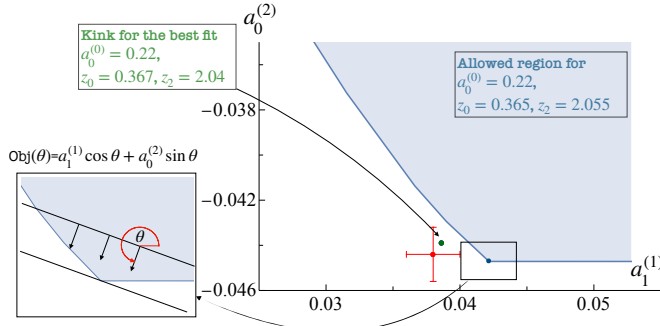

FIG. 3: Examples of kinks in the space of scattering lengths $a_0^{(0)}, a_1^{(1)}$. In blue, we show the kink and corresponding allowed region obtained in [26]. This kink and its allowed region depend on three parameters: $\{a_0^{(0)}, z_0, z_2\}$. In green, we show the position of the kink for the best-fit parameters obtained in this manuscript. The red point and error bars represent the experimental measurement; see also Table II. We observe that while the kink obtained in [26] is inconsistent with the experimental data, the best-fit kink found here is consistent with it. On the left, we represent the target functional for $\theta = 25\pi/18$.

in principle, infinitely many possible choices, not all are equivalent. For our purposes, we found it convenient to adopt the approach developed in [26]. To illustrate why, consider the most general ansatz (3). Assigning specific values to the $S_0$ scattering length[10] $a_0^{(0)}$ and the chiral zeros $z_0$ and $z_2$, we find that the allowed region for the $\{a_1^{(1)}, a_0^{(2)}\}$ scattering lengths exhibits a kink, as shown in figure 3. Changing the values of the $\{a_0^{(0)}, z_0, z_2\}$ parameters, it is possible to move this kink closer to the experimentally measured scattering lengths (the red point in figure 3). When this occurs, the extremal amplitude at the kink closely resembles the pion amplitude.

In this case, the scattering lengths and subleading threshold coefficients align well with experimental data, and the corresponding amplitude contains both the $\sigma$, and the $\rho$ resonances. However, although the $\sigma$ position agrees with the data, the $\rho$ width is larger than observed experimentally.[11] This suggests that by adding the spectral assumption, it may be possible to develop a robust fit ansatz for the pion amplitude.

The Bootstrap problem formulated to construct the fit model for the pion amplitude can be summarized as follows.

---

[6] For simplicity, we do not take into account the error on this parametrization.

[7] It would be interesting to consider the general mixed system of pions and kaons, and obtain the inelasticity as a result of the procedure. See [10, 12, 43] for S-matrix Bootstrap works on mixed-amplitudes.

[8] At tree-level in $\chi$PT: $z_0 = 1/2$, and $z_2 = 2$.

[9] We call the resonances $f_0, f_0', f_0'', \ldots$ to differentiate them. We will use similar notation in other channels. In the $S_0$ channel, we do not include the $\sigma$ in this list.

[10] Scattering lengths are defined from the threshold expansion of the partial amplitudes $\operatorname{Re} t_\ell^{(I)}(s) = 2k^{2\ell}(a_\ell^{(I)} + k^2 b_\ell^{(I)} + \ldots)$, where $k^2 = s/4 - 1$ is the center of mass momentum.

[11] We tried to fine-tune the triplet $\{a_0^{(0)}, z_0, z_2\}$ to match the experimental values of $\sigma$ and $\rho$, but without success. We conjecture that by fine-tuning additional low energy constants it would be possible to generate the $\rho$ in the correct position dynamically.

**Fit Ansatz**

Given $\quad \Theta = \{\theta, a_0^{(0)}, z_0, z_2, m_\rho^2, m_{f_0}^2, m_{f_0'}^2, m_{f_2}^2\}$

Maximize $\quad \mathrm{Obj}(\theta)$
in $\mathcal{A}^{\mathrm{ansatz}}(s,t,u)$

constr. by $\quad t_0^{(0)}(4) = 2a_0^{(0)}, \; t_0^{(0)}(z_0) = 0, \; t_0^{(2)}(z_2) = 0$

$\qquad\qquad S_1^{(1)}(m_\rho^2) = 0, \; S_0^{(0)}(m_{f_0}^2) = 0$

$\qquad\qquad S_0^{(0)}(m_{f_0'}^2) = 0, \; S_2^{(0)}(m_{f_2}^2) = 0$

$s \geq 4 \qquad \mathcal{U}_\ell^{(I)} \succeq 0$ for $\ell \in \mathbb{N}, \; I = 0, 1, 2 \qquad (9)$

The objective we maximize is given by the linear combination

$$\mathrm{Obj}(\theta) = a_1^{(1)} \cos\theta + a_0^{(2)} \sin\theta, \qquad (10)$$

which depends on the angle $\theta$. This objective is a normal functional and is effective at selecting "kinks" [11]. To understand this, note that this objective maximizes the amplitude in the $\{a_0^{(0)}, a_1^{(1)}\}$ space at a point where the tangent vector is orthogonal to $(a_1^{(1)} \cos\theta, a_0^{(2)} \sin\theta)$. At a kink, where multiple tangents exist, many normal functionals naturally converge to the kink point. For $\pi < \theta < 3\pi/2$, this functional span the boundary of the allowed blue region in figure 3. In the left inset, we show a typical choice of angle $\theta$ that will select the kink in the $\{a_0^{(0)}, a_1^{(1)}\}$ space.[12]

The quantities collectively denoted by $\Theta$ are the parameters of the fit and the input for the optimization problem (9). In this problem, four parameters $\theta, a_0^{(0)}, z_0, z_2$ are real, while the mass square are complex. Their real and imaginary parts correspond respectively to the physical mass and width of the resonance. In total, the size of the parameter set in $|\Theta| = 12$. We solve this problem using the standard SDPB solver [46, 47]. The corresponding extremal amplitude depends on the choice of $\Theta$. Next, we will explain how to optimize on $\Theta$.

## III. GRADIENT FREE OPTIMIZATION AND PARTICLE SWARM

For any choice of $\Theta$, the optimal solution of the Bootstrap problem (9) yields a model for the scattering amplitude $\mathcal{A}_\Theta^{\mathrm{ansatz}}(s|t, u)$. To choose $\Theta$, we construct the

$\chi^2(\Theta)$ using the Bootstrap amplitude and the experimental data, and we minimize it. The dependence of the model $\mathcal{A}_\Theta^{\mathrm{ansatz}}(s|t, u)$ is non-linear in $\Theta$, hence we expect the $\chi^2(\Theta)$ to be a non-convex function.

In this Letter, we explore an algorithm especially suited to this class of problems, the Particle Swarm Optimization algorithm (PSO) [48].[13] The PSO is a standard algorithm designed for solving non-convex problems, with a wide range of applications—see [49] for an introduction. The PSO is also gradient-free and does not require the computation of derivatives of the Bootstrap solution in $\Theta$.[14]

We start with $n_p$ particles at random positions $\Theta_0^{(i)}$ to which we assign random velocities $v_0^{(i)}$.

At step $n$, we update their positions following the rule

$$\begin{aligned} v_{n+1}^{(i)} &= \omega v_n^{(i)} + c_1 r_1 (\Theta_n^{(i)} - X_n^{(i)}) + c_2 r_2 (\Theta_n^{(i)} - Y_n), \\ \Theta_{n+1}^{(i)} &= \Theta_n^{(i)} + v_{n+1}^{(i)}. \end{aligned} \qquad (11)$$

The velocity of the $i$-th particle at step $n + 1$ is thus a linear combination of its previous velocity, the distance to its position with the lowest $\chi^2$, denoted by $X_n^{(i)}$ and the distance with the overall best position among all particles $Y_n$. At each step, we first compute the $\chi^2(\Theta_n^{(i)})$ for each particle, then evaluate the various parameters to perform the step.

Three parameters control the algorithm performance: the inertia $\omega$, the cognitive coefficient $c_1$, and the co-operation coefficient $c_2$.[15] The convergence property of the algorithm depends on the choice of these three coefficients.[16] The details of our implementation of the PSO algorithm, which contains an additional adaptive velocity prescription as in [51], can be found in Appendix B.

## IV. BOOTSTRAP FIT RESULTS

We construct the $\chi^2(\Theta)$

$$\chi^2(\Theta) = \sum_{i \in \text{data set}} \left( \frac{\delta_i^{\mathrm{exp}} - \delta_\Theta^{\mathrm{Ansatz}}(s_i)}{\Delta \delta_i^{\mathrm{exp}}} \right)^2 \qquad (12)$$

using experimental and lattice phase shifts $\delta_i^{\mathrm{exp}}$, where the index $i$ stands collectively for the quantum numbers $(I, \ell) = \{(0, 0), (1, 1), (2, 0), (0, 2)\}$, and the energy $s_i$ of the measurement. Our ansatz for the phase shift is obtained by projecting $\mathcal{A}_\Theta^{\mathrm{ansatz}}$ in partial waves and using the definition $S_\Theta = |S_\Theta| \exp(2i\delta_\Theta)$.

---

[12] We could consider a different objective $\mathrm{Obj}'(\theta, \phi) = a_0^{(0)} \cos\phi \sin\theta + a_0^{(2)} \sin\phi \sin\theta + a_1^{(1)} \cos\theta$, and replace the $a_0^{(0)}$ parameter in (9) with a new angle $\phi$. This would lead to an equivalent formulation of the Bootstrap ansatz (9). From this view, it is evident we are moving along the two-dimensional boundary of the three-dimensional parameter space of the scattering lengths. This can be generalized by adding an arbitrary number of low energy observables, which amounts to scanning over a larger space of amplitudes.

[13] We are grateful to Balt van Rees for pointing out this method.

[14] As explained in [50], it is possible to efficiently compute the gradient of a Bootstrap problem solved with the interior point method as the one implemented in SDPB.

[15] The $r_{1,2}$ are random numbers uniformly distributed between $[0, 1]$, and are drawn at each step.

[16] In other versions of the algorithm, those parameters can be promoted to be dynamical.

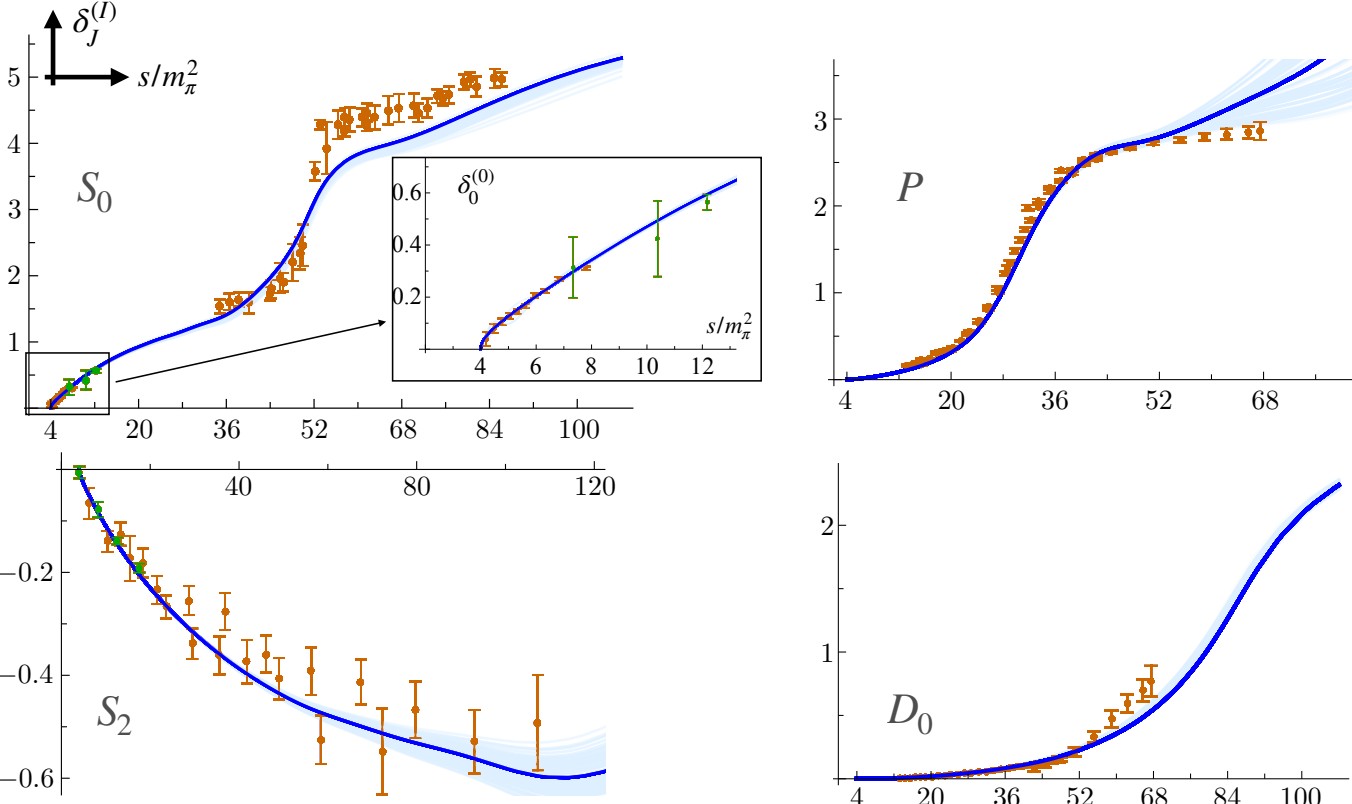

FIG. 4: The four channels used to fit the pion amplitude. The points respectively in orange and green are experimental and lattice data. The thick blue curve is the best fit, the light blue cloud is given by all the curves with sub-optimal $\chi^2$.
(Notation: $S_0$ stands for the $(I, \ell) = (0, 0)$ channel, $P$ for the $(1, 1)$, $S_2$ for the $(2, 0)$, and $D_0$ for the $(0, 2)$.

The input data used in (12) are from [52–56], except for the $S_0$ wave. The data in the $S_0$ channel above $s \approx 20$ extracted from the old experiments are often incompatible and suffer from unknown systematic errors. Following [57], we used only a selection of data points.[17] Close to the threshold, the experimental situation was cleared by the CERN experiment NA48/2 [58]. We also include lattice data from the RBC and UKQCD collaborations in the $S_0$ and $S_2$ channels extrapolated at the physical pion mass [59].[18]

The result of the Bootstrap fit is shown in figure 4. The orange data points are experimental, the green are taken from the lattice. The dark blue curve is the best fit with minimum $\chi^2(\Theta) \approx 40$. We do not assign a statistical meaning to the value of the $\chi^2$, but take it as a likelihood measure. The largest contribution to the $\chi^2$ comes from the $P$ wave, with $\chi^2 \approx 30$, where the experimental error is almost negligible. We plan to investigate this channel with more care in the future, replacing the phase shift

data with the determination from the pion form factors [60, 61], and with lattice extrapolations [62]. The second largest contribution of order ten is due to the $S_0$ wave. Here we observe a systematic error in reproducing the phase above the $K\bar{K}$ threshold $s \approx 50$ (the $\chi^2$ below this energy is of order one). We believe this is due to the lack of an explicit threshold in our ansatz at that scale, as corroborated by the slower convergence of the Bootstrap problem, see also Appendix A. Both $S_2$ and $D_0$ waves have $\chi^2$ of order one.

We run the PSO algorithm for $N_{\text{iter}} = 60$ steps, using $n_p = 10$ particles. We explore 600 points in the $\Theta$ parameter space and at each point, we construct the full analytic amplitude solving the Bootstrap problem (9). To estimate errors we consider a subset of amplitudes, and compute the weighted average using the value of the $\chi^2$. We observe that 50% of our set have $\chi^2 < 100$, and 15% have $\chi^2 < 50$. We compare the errors estimated using these two cutoffs and find that those obtained with the looser one are a factor of two larger. The light-blue curves in figure 4 are drawn from amplitudes with $\chi^2(\Theta) < 50$. In the remainder of this paper, we will follow the same color scheme.

In the inset in figure 4, we zoom in on the NA48/2 and lattice data close to the threshold. The point at the kaon mass $s \approx 12$ is from lattice [59]. This point is crucial to

---

[17] We are grateful to Jose Ramon Pelaez for highlighting this point.
[18] For the lattice data points, the errors on the phase shift and the energy are strongly correlated. We do not include this effect in this work. In this case, we use the isospin symmetric value of the pion mass $m_\pi = 135$ MeV to express energies in dimensionless units.

help stabilize the amplitude between the threshold and the cluster of points around $s \approx 40$.

| $\Theta$ | Bootstrap Fit | Literature |
|---|---|---|
| $a_0^{(0)}$ | $0.217 \pm 0.002$ | $0.2196 \pm 0.0034$ [58] |
| $z_0$ | $0.368 \pm 0.008$ | |
| $z_2$ | $2.040 \pm 0.004$ | |
| $m_\rho$ | $(5.546 \pm 0.005)$ $+i(0.538 \pm 0.002)$ | $(5.555 \pm 0.015)$ $+i(0.528 \pm 0.013)$ |
| $m_{f_0}$ | $(7.18 \pm 0.04)+i(0.26 \pm 0.02)$ | $(7.25 \pm 0.11)+i(0.21 \pm 0.07)$ |
| $m_{f_0'}$ | $(9.8 \pm 0.2)+i(1.7 \pm 0.1)$ | $(9.8 \pm 0.7)+i(1.3 \pm 0.9)$ |
| $m_{f_2}$ | $(9.26 \pm 0.03)+i(0.69 \pm 0.04)$ | $(9.26 \pm 0.08)+i(0.73 \pm 0.08)$ |
| $\theta/\pi$ | $1.328 \pm 0.026$ | |

TABLE I: Our estimate of the fit parameter compared and corresponding values quoted in the literature. Experimental values of the resonances are given by the PDG average [5].

In Table I, we list our estimate for the fit parameters and the corresponding errors. The error in this table is conservative and estimated using all the points with $\chi^2(\Theta) < 100$ corresponding to half of the whole dataset of amplitudes produced.

Beyond the statistical error, there are three more sources of systematics. The first concerns the convergence of the Bootstrap model $\mathcal{A}_\Theta^{\mathrm{ansatz}}(s|t,u)$ in equation (3) that depends on $N_{\mathrm{vars}}$ the number of free variables $\alpha_{n,m}^{(\sigma)}$, and $\beta_{n,m}^{(\sigma)}$. We used two ansatzes respectively with $N_{\mathrm{vars}} = 397$ and $N_{\mathrm{vars}} = 547$. All the results we quote in this paper are taken from the latter. [19]. The difference between the fit parameters estimated from the two ansatzes is comparable with the statistical error unless stated otherwise. The second source comes from the PSO search algorithm, which does not guarantee finding the global minimum. We checked the stability of our fit by performing different searches varying the search parameters that produced almost the same result. The last source of systematics comes from the choice of the Bootstrap functional. In this case, we have tested an alternative Bootstrap formulation obtaining the same fit.

## V. PREDICTIONS

The advantage of our strategy is that the result of the fit is a full analytic amplitude. Next, we study its properties away from the region directly constrained by the fit.

---

[19]To demonstrate the numerical robustness of our physical predictions, we performed more runs using different foliation parameters. The detailed parameters and results are summarized in C

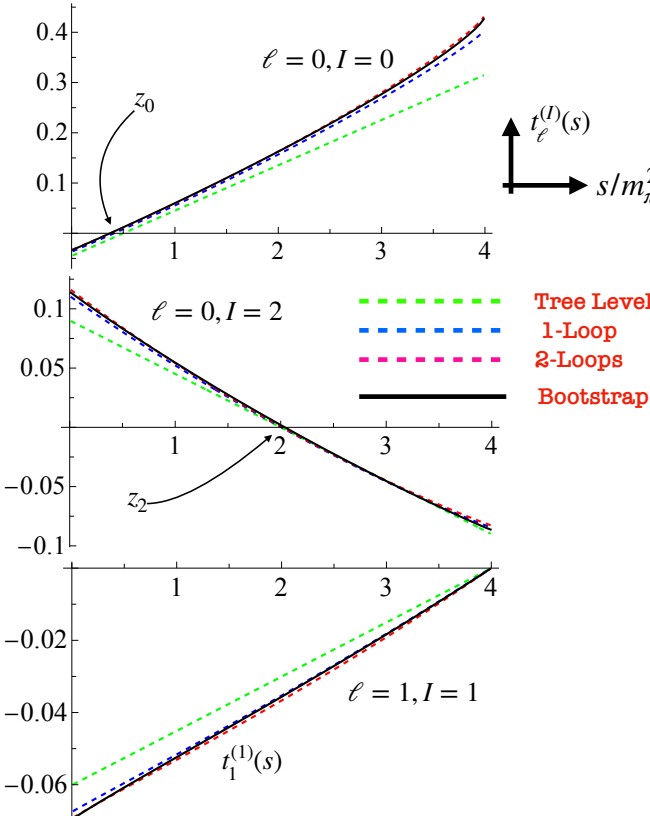

FIG. 5: Partial wave amplitudes $t_\ell^{(I)}$ in the real strip $0 < s < 4$. The black line is the Bootstrap prediction, dashed lines are different approximations from $\chi$PT [63].

### A. Amplitude below threshold

The $\chi$PT expansion is a reliable approximation of the pion amplitude at low energy. In figure 5, we plot the partial waves $t_0^{(0)}$, $t_0^{(2)}$, and $t_1^{(1)}$ of our best fit (the solid black line) in the sub-threshold region $0 < s < 4$, and we compare it with perturbation theory [63]. The tree level, denoted with the dashed green line is given by

$$t_0^{(0)} = \frac{2s-1}{32\pi f_\pi^2}, \quad t_1^{(1)} = \frac{s-4}{96\pi f_\pi^2}, \quad t_0^{(2)} = \frac{2-s}{16\pi f_\pi^2}, \quad (13)$$

where $f_\pi$ is the pion decay constant. The higher loop expressions are more involved and are also plotted in figure 5. The two chiral zeros $z_0$, and $z_2$ are also visible in the figure. We emphasize that the only input in this region is the existence of the two chiral zeros, not their position.

Next, we extract the threshold parameters from our amplitude and compare them with the values of [64] in Table II. We find a very nice agreement for the leading and sub-leading threshold coefficients for all waves below $\ell = 2$. The scattering lengths we extract are different for higher spins but of the same order. It would be interesting to investigate the reason for this discrepancy.

| | Bootstrap Fit | Literature |
|---|---|---|
| $a_0^{(2)}$ | $(-0.432 \pm 0.001) \times 10^{-1}$ | $(-0.444 \pm 0.012) \times 10^{-1}$ |
| $a_1^{(1)}$ | $(0.380 \pm 0.002) \times 10^{-1}$ | $(0.379 \pm 0.05) \times 10^{-1}$ |
| $b_0^{(0)}$ | $0.265 \pm 0.030$ | $0.276 \pm 0.006$ |
| $b_0^{(2)}$ | $(-0.797 \pm 0.002) \times 10^{-1}$ | $(-0.803 \pm 0.012) \times 10^{-1}$ |
| $b_1^{(1)}$ | $(0.61 \pm 0.02) \times 10^{-2}$ | $(0.57 \pm 0.01) \times 10^{-2}$ |
| $a_2^{(0)}$ | $(0.53 \pm 0.11) \times 10^{-2}$ | $(0.175 \pm 0.003) \times 10^{-2}$ |
| $a_2^{(2)}$ | $(0.51 \pm 0.18) \times 10^{-3}$ | $(0.170 \pm 0.013) \times 10^{-3}$ |
| $a_3^{(1)}$ | $(1.5 \pm 0.4) \times 10^{-4}$ | $(0.56 \pm 0.02) \times 10^{-4}$ |

TABLE II: Except for the values of $a_0^{(2)}$ taken from [58], all other threshold parameters are taken from [64].

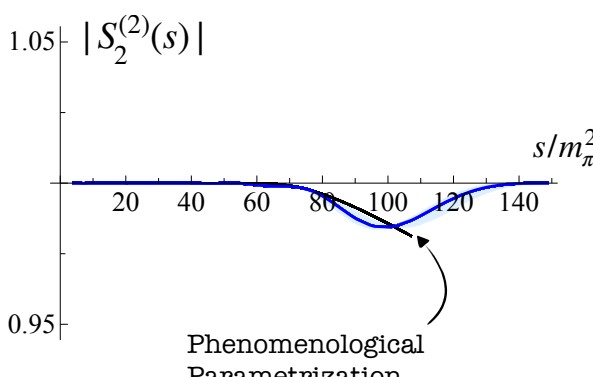

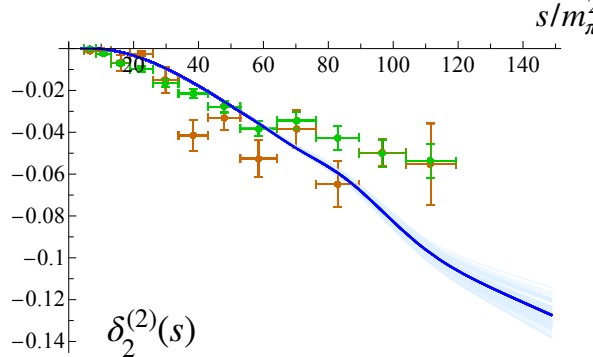

FIG. 6: Top panel: Elasticity of the $D_2$ wave as a function of $s/m_\pi^2$. The black line represents the parametrization from [57]. Bottom panel: Phase shift of the $D_2$ wave compared with the available experimental data. The green and orange points correspond to the two possible phase determinations reported in [56].

### B. The $D_2$ partial wave

In figure 6, we plot the elasticity $|S_2^{(2)}|$ and the $\delta_2^{(2)}$ phase shift prediction. The color scheme follows the one defined in figure 4. The data for the phase shift are taken from [56], orange and green correspond to two possible determinations. We compare the elasticity profile with the phenomenological parametrization from [57]. It is interesting to notice a dip in the unitarity around $s \approx 90$. The dip is insensitive to Bootstrap systematics. We think

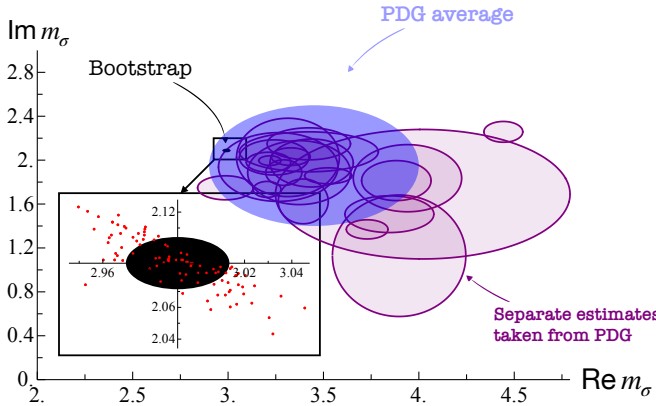

FIG. 7: Determinations of real $\mathrm{Re}\, m_\sigma$ and imaginary part $\mathrm{Im}\, m_\sigma$ of the mass of the $\sigma$. The black ellipse highlighted in the inset is our Bootstrap determination.

that the emergent particle production in this channel is a consequence of the inelasticity profile injected in the $D_0$ wave. This effect is compatible with our expectation that including mixed amplitudes with kaons and pions might lead to a correct prediction for the inelasticity.

### C. Spectrum

Beyond the spectrum assumed and fitted using the experimental data, we also observe several zeros dynamically generated in our construction.

We begin by discussing the $f_0(500)$ resonance, commonly referred to as the $\sigma$. We summarize the phenomenological determinations of its position in figure 7. The pink ellipses denote all the estimates of its complex mass reported in the PDG since 2001 [5]. The blue ellipse is the PDG average. Our estimate is represented by the black ellipse highlighted in the inset.

To estimate the position of the $\sigma$ we look for a zero at low energy in the $S_0^{(0)}$ $S$-matrix using the Newton method. We repeat the operations for all amplitudes with $\chi^2(\Theta) < 50$, and perform a weighted average to predict the mass and determine the error. The red dots in the inset are the individual determinations from this sample. Our final estimate is reported in Table III. In figure 8, we also show the density plot for $|S_0^{(0)}|$ in the upper-half complex $s$ plane. The left-most zero is the $\sigma$. In the same plot, we observe two additional resonances that we identify as the $f_0(980)$, and $f_0(1370)$.

Our $\sigma$ determination is shifted away from the center of the PDG average. This is correlated with the lattice point at $s = m_K^2 \approx 12$ which was not used in previous studies, and impacts the growth of the phase in the $S_0$ wave.

We also find an additional state in the $P$-wave. The density plot for the $|S_1^{(1)}|$ in the upper half $s$ plane is shown in figure 9 for the best-fit amplitude. The two black dots are zeros of $S_1^{(1)}$. The left-most zero is iden-

| | Prediction | PDG average |
|---|---|---|
| $m_\sigma$ | $(2.99\pm0.02)+i(2.09\pm0.02)$ | $(4.3\pm1.5)+i(3.3\pm2.6)$ |
| $m_{\rho'}$ | $(9.5\pm0.6)+i(0.55\pm0.05)$ | $(10.7\pm0.3)+i(1.45\pm0.05)$ |
| $m_{\rho_3}$ | $(16.7\pm0.6)+i(0.7\pm0.2)$ | $(12.3\pm0.1)+i(0.68\pm0.05)$ |
| $m_{\mathbb{T}}$ | $(14.6\pm0.3)+i(2.4\pm0.1)$ | ? |

TABLE III: Experimental values of the resonances are taken from [5].

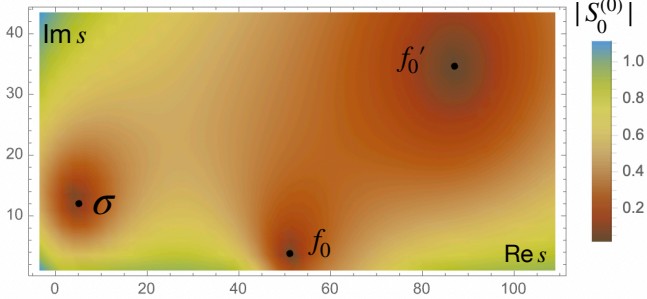

FIG. 8: $|S_0^{(0)}|$ in the upper half complex $s$ plane plotted for the best-fit amplitude. We highlight with black dots the three scalar resonances found in this channel.

tified as the $\rho(770)$ resonance. Its position is fixed by the fitting procedure. The second black dot on the right is dynamically generated and has a mass 5% lower than the experimental determination of the $\rho(1450)$. However, its width is half of the experimentally measured value. This resonance still suffers from systematics: we do not have the inelasticity profile at this energy, and $N_{\text{vars}}$ of the Bootstrap problem is not large enough to accurately describe this region. We suspect that dealing with this systematics might lead to the correct width in our procedure.

For higher spins, our amplitude dynamically generates a spin $\ell = 3$ resonance, this is the first sign of Reggeization. This is typical in the amplitudes constructed using the S-matrix Bootstrap where the high energy behavior instead of being erratic is physical, although it converges slowly, see also section V E. The mass of the $\rho_3$ obtained is 20% above the experimental value, and its width is larger. This resonance, however, is still affected by the convergence of the Bootstrap ansatz, and improving the numerics might lead to a better agreement.[20]

### D. The tetraquark

As already indicated in the introduction, examining the $S_2$ wave reveals an unexpected result. Continuing

--------

[20]The other possibility is that to obtain the precise Regge trajectory we might need additional fine-tuning, which can be realized by enlarging the set of fit parameters or generalizing the Bootstrap problem we use to construct the amplitude.

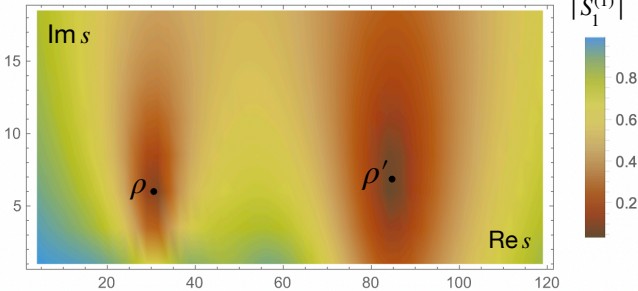

FIG. 9: $|S_1^{(1)}|$ density profile in the upper half complex $s$ plane. We highlight with black dots the two $\rho$ resonances in our amplitude.

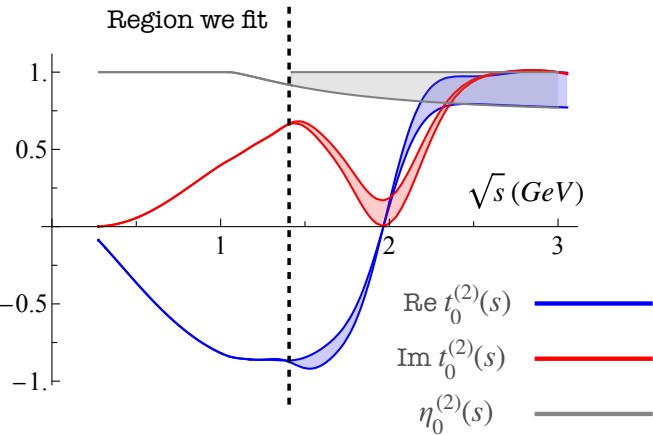

FIG. 10: Real and imaginary parts of the amplitude in the $S_0^{(2)}$ channel. The error bars indicate the systematic uncertainties in the line shape of this function arising from the unknown inelasticity above 1.4 GeV. This error is comparable to the statistical error of our fit reported in figure 2.

the phase shift $\delta_0^{(2)}$ to high energy, as shown in figure 2, we observe a broad resonance with isospin $I = 2$. This indicates the presence of a genuine Tetraquark state [6, 7]. In the physical world, where electric charge is reintroduced, such particles would carry a charge of two. Its mass and width in physical units are respectively 2 GeV and 600 MeV. It would be intriguing to examine the invariant mass distributions of $\pi^\pm\pi^\pm$ in $B$-meson decays that provide sufficient energy to reach the 2 GeV scale. Examples include the decay $B^+ \to \pi^-\pi^+\pi^+$ [65], which can be studied at LHCb, as well as similar decays where the $\pi^-$ is replaced by either $D^-$ or $K^-$.

We determine the mass and width of the Tetraquark by locating the zero in the complex $s$ plane in the $S_2$ wave. This determination is free from Bootstrap-related systematics, as the amplitude in this channel is well-converged. However, we have not accounted for inelastic effects in this region, which could slightly alter the resonance's position.

In figure 10, we plot the real and imaginary parts of

the partial wave amplitude $t_0^{(2)}$ as a function of $\sqrt{s}$ measured in GeV. From an amplitude perspective, this represents the experimental signal expected in the presence of a Tetraquark.[21] The black dashed line separates the region where we fit the data and incorporates the inelasticity profile from the high-energy region, which is unconstrained. The gray curve represents the elasticity profile $\eta_0^{(2)}(s)$ taken from [42]. Above $\sqrt{s} = 1.4$ GeV, we set $\eta_0^{(2)}(s) = 1$ in our procedure. For illustrative purposes, we show the amplitude profile using the $S_2$ wave parametrization $S_0^{(2)}(s) = |S_0^{(2)}(s)| \exp(2i\delta_0^{(2)}(s))$, where $\delta_0^{(2)}$ is the phase obtained from our best fit, and $|S_0^{(2)}(s)|$ is a generic function bounded by $\eta_0^{(2)}(s) \leq |S_0^{(2)}(s)| \leq 1$, for $s > 1.4$ GeV. The lower bound corresponds to the extrapolation of [42] to higher energies, as shown by the gray area in figure 10. The blue and red bands depict how uncertainty in the elasticity function influences the shape of the amplitude.

It would be interesting to understand the mechanism behind the emergence of the Tetraquark, using, for instance, the Roy equations [3]. Notably, there is a two-dimensional example that qualitatively mirrors this scenario: the two-dimensional theory of the QCD flux tube. In this context, the QCD $\rho$ is analogous to the axion, while the $\sigma$ corresponds to the dilaton. The analogy is striking, as this model features a sharp axion and a broad dilaton [22]. Furthermore, in [66], it was observed that crossing symmetry necessitates a broad resonance in the symmetric channel. A similar mechanism might underlie the presence of the Tetraquark identified in our amplitude.
[22]

### E. High Energy

Finally, we examine the Regge behavior of our amplitude and compare it with experimental data.

One of the most intriguing recent insights into S-matrix Bootstrap amplitudes is their emergent high-energy behavior. Although the Bootstrap ansatz in (3) approaches a constant at infinity, there exists an intermediate regime where the numerical amplitudes align with Regge theory. Specifically, at large $s$ and fixed $t$, the amplitudes exhibit the expected behavior $\mathcal{A} \sim s^{\alpha(t)}$ [24, 77–79].

In figure 11, we compare the experimental total cross sections for $\pi^+\pi^-$ and $\pi^-\pi^-$ with those extracted from our best-fit amplitude, shown in blue[23]

$$\sigma_{\pi^+\pi^-}(s) = \left.\frac{\mathrm{Im}\left[\mathcal{A}(s|t,u) + \mathcal{A}(t|s,u)\right]}{\sqrt{s(s-4)}}\right|_{t=0},$$
$$\sigma_{\pi^-\pi^-}(s) = \left.\frac{\mathrm{Im}\left[\mathcal{A}(t|s,u) + \mathcal{A}(u|s,t)\right]}{\sqrt{s(s-4)}}\right|_{t=0}, \quad (14)$$

The solid blue curve represents our fit with the highest $N_{\mathrm{vars}} = 547$, while the light blue curve corresponds to the best fit with $N_{\mathrm{vars}} = 397$.

In $\sigma_{\pi^+\pi^-}$ we observe distinct peak structures corresponding to various resonances in the spectrum, with the first and most prominent being the $\rho$ peak. In contrast, $\sigma_{\pi^-\pi^-}$ is not expected to exhibit resonance-related peaks, except for the Tetraquark. At asymptotically high energies, we expect a constant or slightly growing behavior consistent with the pomeron exchange (see for instance [80]). As $N_{\mathrm{vars}}$ increases, we expect the Bootstrap amplitude will better approximate this behavior, which is not explicitly enforced in our ansatz (3).

## VI. DISCUSSION AND OUTLOOK

This Letter introduces a novel constrained approach for fitting experimental data. The key advantage of our method lies in its control over theoretical approximations. The amplitude model employed is a genuinely analytic, crossing-symmetric function that satisfies non-perturbative unitarity. The algorithm combines semi-definite programming and non-convex optimization techniques: for the Bootstrap component, we use SDPB [46, 47], while for the particle swarm optimization (PSO), we implemented a straightforward Mathematica notebook.

The constructed $\pi\pi \to \pi\pi$ amplitude exhibits several remarkable features. It reproduces the experimental data, accurately predicts the $G$-parity plus spectrum below 1.4 GeV, aligns with $\chi$PT at low energies, and qualitatively matches the expected behavior in the high-energy regime. Additionally, the model predicts a resonance heavier than the $\rho$ in the $P$ wave, with a mass comparable to the real-world $\rho(1450)$, as well as a spin-three resonance approximately 20% heavier than the experimental value. The $D_2$ phase shift and inelasticity also show good agreement with experimental data and phenomenological models. In figure 12 we present an overview of the bootstrap fit procedure, summarizing the key input data and the corresponding output results.

Interestingly, our analysis predicts a tetraquark state with a mass of approximately 2 GeV in the $I = 2$ channel. This is a fully non-perturbative prediction that, to

---

[21] We thank Alessandro Pilloni for providing insights into the input required for the experimental data analysis.

[22] Following the completion of this work, Luiz Vale Silva drew our attention to the data presented in [67]. Interestingly, the change in the phase behavior predicted by the Bootstrap and shown in Figure 2 is consistent with the findings reported in [67]. The mechanism underlying this behavior has been studied in [68, 69]. We are grateful to Luiz Vale Silva for bringing these references to our attention.

---

[23] In isospin components $\sigma_{\pi^+\pi^-} \propto \frac{1}{6}(2\mathcal{T}^{(0)} + 3\mathcal{T}^{(1)} + \mathcal{T}^{(2)})$, and $\sigma_{\pi^-\pi^-} \propto \mathcal{T}^{(2)}$.

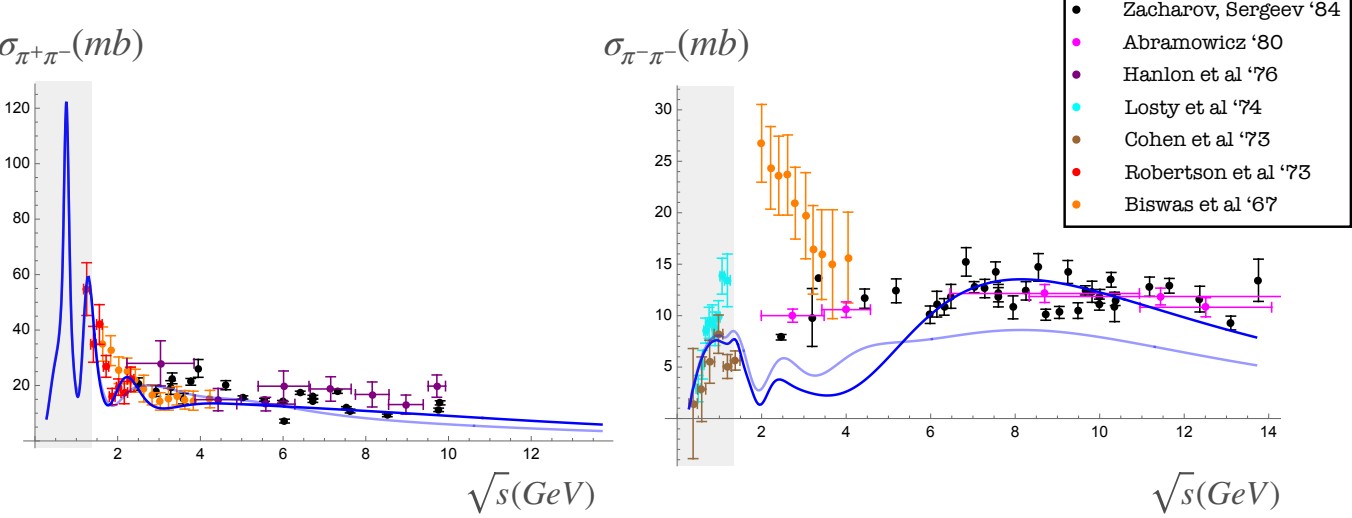

FIG. 11: Total cross sections for $\pi^+\pi^- \to \pi^+\pi^-$ and $\pi^-\pi^- \to \pi^-\pi^-$. The gray shaded area is the region where we fit the data. The blue curve is the best fit for $N_{\text{vars}} = 547$, in light blue the best fit for $N_{\text{vars}} = 397$. Above 2 GeV there is still room to improve numerics, but there is already a very good agreement between Bootstrap and the experimental data [55, 70–76].

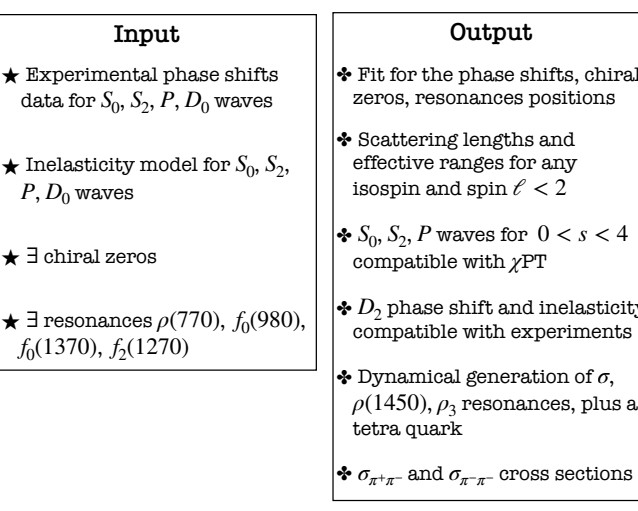

FIG. 12: Summary of the bootstrap fit procedure.

our knowledge, has not been proposed in previous models. Given that our constructed amplitude accurately describes multiple key features of $\pi\pi$ scattering, this state presents an ideal candidate for experimental verification, particularly in $B$-meson decays at LHCb.

**Extensions and Numerical Refinements**

Several enhancements could further improve the pion amplitude model.

1. Incorporating Lattice QCD Data more consistently including the correlation matrix for the lattice data in the $S_0$ wave from [59]. This would help refine the determination of the $f_0(500)$ mass parameters,

particularly in resolving the small tension of our determination with previous Roy equation analyses [81].

2. Using the $P$ wave phase shift $\delta_1^{(1)}$ extracted from form factors [60, 61] and from lattice QCD extrapolations at the physical pion mass [62] might improve our determination of the $\rho$ resonance pole parameters.

3. Improving the description of the inelasticity profile by including additional scattering processes, such as $K\bar{K}$ scattering and mixed meson channels, would increase the self-consistency of the fit.

4. Refining the Numerical Optimization. The current non-convex optimization procedure can be improved using alternative numerical techniques, including hybrid gradient-based methods [82] and machine learning-assisted approaches.

5. Generalizing the Functional Form. The objective functional in Eq. (10) was constructed based on heuristic criteria. Exploring alternative formulations including higher-order low-energy constants in the objective as done in previous S-matrix Bootstrap studies [16, 17, 19, 20, 27, 29, 83] could provide a more systematic way to generate self-consistent amplitudes.

6. Adding Form Factors [13, 23, 30, 31, 84]. Extending the system of QFT correlators including form factors and spectral densities with different chiralities will allow us to extract the values of QCD condensates and to fit vector and axial spectral densities.

This will give access to axial resonances, without the inclusion of mixed amplitudes.[24]

### Experimental Implications and Broader Applications

The predicted doubly charged tetraquark resonance at 2 GeV provides an exciting opportunity for direct experimental confirmation in $B$ decays. Future studies could focus on dedicated searches in high-energy experiments such as LHCb, BESIII, and Belle II, where doubly-charged hadronic final states could serve as potential signatures of this state. The Bootstrap prediction of the amplitude signal in figure 10 might facilitate the detection of this state and help the analysis of the experimental data.

Beyond QCD, the Bootstrap Fit methodology may have broader applications in strongly coupled field theories. Extending this approach to other two-body scattering processes – such as nucleon-nucleon interactions, Glueball-Glueball scattering [21], or even classical gravitational amplitudes used in the extraction of waveforms – could provide new insights into the universal structure of non-perturbative quantum field theories.

### Conclusion

Our results demonstrate that integrating bootstrap constraints with direct experimental fits provides a powerful, model-independent framework for extracting hadronic properties. Unlike traditional dispersion relation methods, our approach fully incorporates analyticity, unitarity, and crossing symmetry at all energy scales. This makes it particularly well suited for investigating strongly interacting theories such as QCD, or strongly coupled UV completions of the standard model.

Moving forward, an important challenge is to extend our approach toward rigorous bounds on resonance positions in the complex $s$-plane, similar to the methodology employed in [86] for the Ising Field Theory in 1+1 dimensions. Further, constructing amplitudes that dynamically generate all resonances without explicit spectral assumptions remains an open problem worth investigating.

Ultimately, our study highlights the potential of the S-matrix Bootstrap not only as a tool for theoretical constraints but also as a practical means for extracting real-world physical observables in a controlled, non-perturbative manner. The continued refinement of this methodology could lead to significant advancements in the study of strongly coupled quantum field theories across multiple disciplines.

—————

[24]Form factors computed using other methods, such as in [85], where Hamiltonian truncation was employed, could also be used to constrain the amplitude.

### ACKNOWLEDGEMENTS

We thank Mattia Bruno, Gilberto Colangelo, Miguel Correia, Barak Gabai, Victor Gorbenko, David Gross, Aditya Hebbar, Denis Karateev, Luciano Maiani, Harish Murali, Jose Ramon Pelaez, Joao Penedones, Alessandro Pilloni, Jiaxin Qiao, Riccardo Rattazzi, Slava Rychkov, Balt van Rees, Luiz Vale Silva, David Simmons-Duffin, Pedro Vieira, and Alexander Zhiboedov for useful discussions.

We thank Gilberto Colangelo, Joao Penedones, Alessandro Pilloni, Balt van Rees, Pedro Vieira, Mark Wise and Alexander Zhiboedov for the many useful comments on the draft.

Research at the Perimeter Institute is supported in part by the Government of Canada through NSERC and by the Province of Ontario through MRI. AG is supported by the European Union - NextGenerationEU, under the programme Seal of Excellence@UNIPD, project acronym CluEs. The work of KH is supported by the Simons Foundation grant 488649 (Simons Collaboration on the Nonperturbative Bootstrap), by the Swiss National Science Foundation through the project 200020 197160, through the National Centre of Competence in Research SwissMAP and by the Simons Collaboration on Celestial Holography. NS's work is supported by Simons Foundation grant 488657 (Simons Collaboration on the Nonperturbative Bootstrap). This material is based upon work supported by the U.S. Department of Energy, Office of Science, Office of High Energy Physics, under Award Number DE-SC0011632. This material is based upon work supported by the U.S. Department of Energy, Office of Science, Office of High Energy Physics, under Award Number DE-SC0011632.

### Appendix A: Details of the S-matrix bootstrap numerics

The ansatz used to produce the results of this letter has multi-foliation $\Sigma = \{20/3, 30, 50, 86\}$. As discussed in [22], to best approximate resonant structures it is convenient to set the centers of the foliations around their expected positions. That explains our choice for the set $\Sigma$ which is tuned on the expected position of the $\rho(770)$ with $m_\rho^2 \approx 30$, the $f_0(980)$ with $m_{f_0}^2 \approx 50$, and the $f_2(1260)$ with $m_{f_2}^2 \approx 86$.

Let us call $N = N_{20/3}$, and $M = N_\sigma$, when $\sigma$ is any other foliation. To produce the plots in the main text we used $N = 14$, and $M = 12$ for a total of $N_{\text{vars}} = 547$ variables. We have also performed a second run with $N = 12$, and $M = 10$ for a total of $N_{\text{vars}} = 397$ to check the systematic due to the size of the ansatz.

To impose the unitarity constraints we project numerically our ansatz into partial waves. We use a union of grids, one for each foliation. The grid is defined through

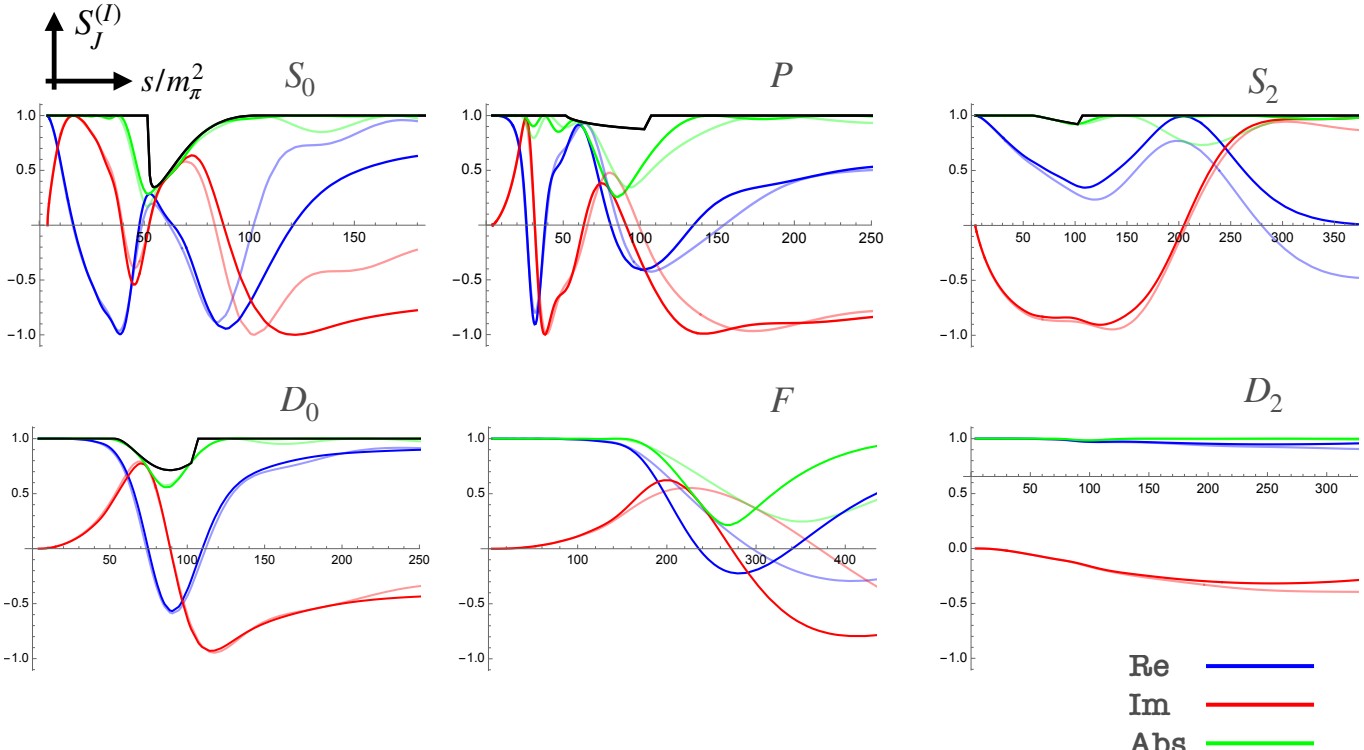

FIG. 13: Real (blue), imaginary (red), and absolute value (green) of the phase shifts for various channels. Thick lines are from our best numerics with $(N, M) = (14, 12)$, light lines for $(N, M) = (12, 10)$. The elasticity profiles used as a constraint are denoted in black. We do not draw it for the channels where we impose the simple probability conservation $|S_J^{(I)}| \leq 1$.

the map

$$s(\phi) = \frac{\sigma - (8 - \sigma)\cos\phi}{1 + \cos\phi} \tag{A1}$$

where $0 \leq \phi \leq \pi$ is the upper boundary of the unit disk. We finally discretize $\phi$ on the roots of the Chebyschev polynomials

$$\phi_k = \frac{\pi}{2}\left(1 + \cos\frac{\pi k}{N_{\text{points}} + 1}\right), \tag{A2}$$

where $k = 1, \ldots, N_{\text{points}}$. The maximum number of points used is $N_{\text{points}} = 300$ for $\sigma = 20/3$, and $N_{\text{points}} = 150$ for all other foliations. We have performed numerical tests using grids with different numbers of points finding no significant dependence on it.

We have projected our ansatz onto partial waves (7) up to spin $\ell = 12$, and run our numerics with both maximum spin $L_{\max} = 10, 12$. The reason that we can keep $L_{\max}$ so low is due to the addition of improved positivity constraints [18]

$$\text{Im}\,\mathcal{T}^{(I)}(s, t) - 16\pi \sum_{\ell=0}^{L_{\max}} (2\ell + 1)P_\ell(1 + 2\tfrac{t}{s-4})\text{Im}\,t_\ell^{(I)}(s) \geq 0 \tag{A3}$$

for $0 \leq t < 4$, $s > 4$, and $I = 0, 1, 2$. By imposing this condition we constrain the tail of spins higher than $L_{\max}$ that we do not explicitly bound with numerical unitarity.

In figure 13 we show the partial S-matrices $S_J^{(I)}$ obtained using two ansatzes respectively with $(N, M) = (14, 2)$ (dark lines), $(N, M) = (12, 10)$ (light lines). We plot their real (blue), imaginary (red), and absolute values (green). The black solid line is the inelasticity profile due to $\pi\pi \to K\bar{K}$ taken from [42] (for the $S_0$ channel we assume the big-dip scenario) that we impose up to $s \approx 100$. Above that energy we keep the inequality $|S_J^{(I)}| \leq 1$.

The unitarity constraints are sufficiently saturated in all these channels except for the region around $s \approx 100$ in the P wave around the position of the $\rho'$, and in the F wave around $s \approx 250$ at the position of the $\rho_3$. These dips in unitarity around resonances are a typical numerical artifact of our finite $N_{\text{vars}}$ truncation. It would be interesting to increase both $N$ and $M$ to higher values and see if the positions of these particles better align with their experimental determinations.

## Appendix B: Particle Swarm implementation

Here we describe our implementation of the PSO algorithm in detail.

To begin the algorithm, we first choose a search region in the fit parameter space where we will look for the minimum of the $\chi^2(\Theta)$ function. Specifically, we select

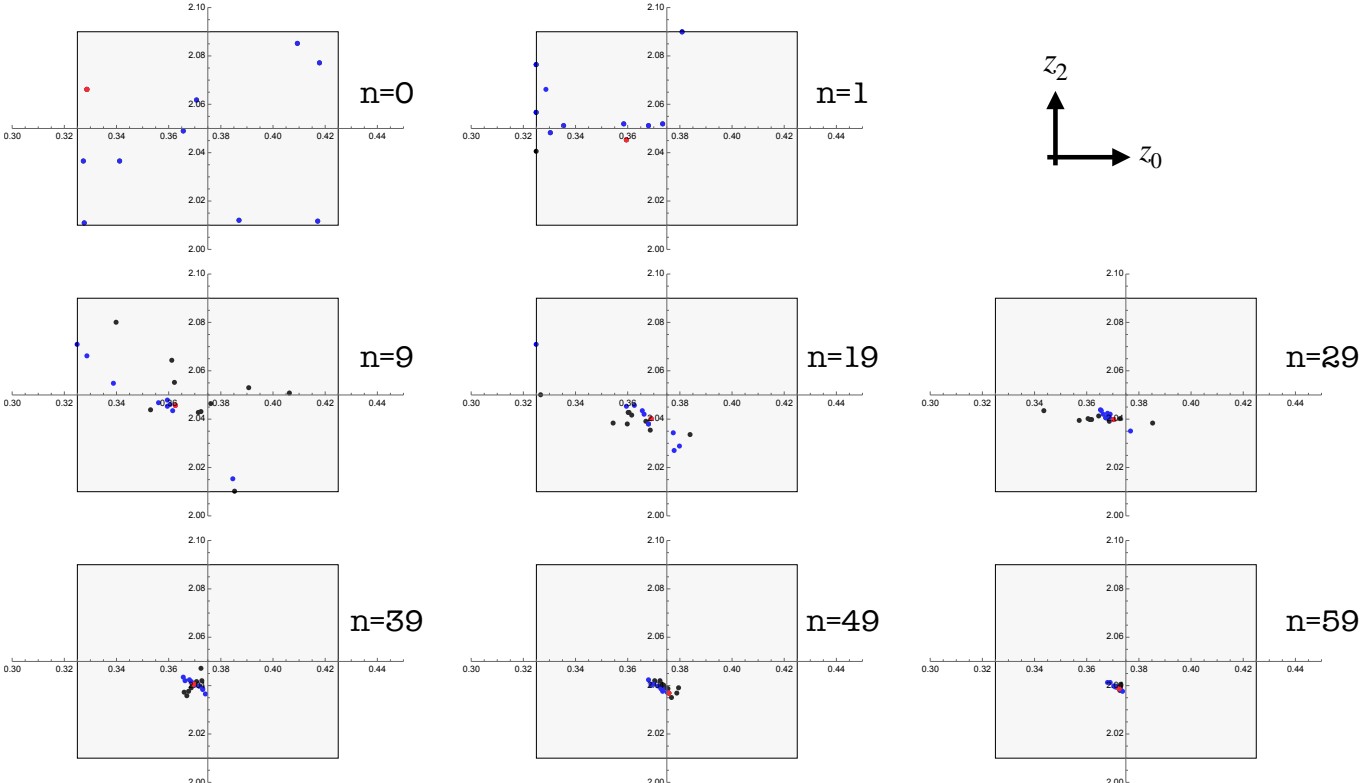

FIG. 14: History of the PSO search in the plane $(z_0, z_2)$. At each step $n$, we plot in black the positions of the particles and highlight the best individualistic positions $X_n^{(i)}$ in blue, and the collective best position $Y_n$ in red. As $n$ increases the particles slowly concentrate around the optimal region. The complete movie can be seen in
`https://giphy.com/gifs/Q5mtDePqPtfSqaQI2z`.

a region $\vec{\Theta}_{\min} \leq \vec{\Theta} \leq \vec{\Theta}_{\max}$ centered around the phenomenological estimates of the fit parameters.

We set the number of particles to $n_{\mathrm{p}} = 10$. This number represents the size of the "swarm". The initial positions of the particles are generated randomly, with values drawn from a Gaussian distribution centered at the region's center, and a variance proportional to its size, denoted by $\vec{\epsilon} = \frac{1}{2}(\vec{\Theta}_{\max} - \vec{\Theta}_{\min})$. Similarly, the initial velocities of the particles are sampled from a Gaussian distribution with zero mean and variance $\vec{\epsilon}$.

We start the algorithm by updating the positions and velocities of the particles according to equation (11). Additionally, if the particle $i$ moves outside the boundaries of the box during the update step along some direction $j$, we then place it back on the boundary setting $(\Theta_{n+1}^{(i)})_j = (\Theta_{\max/\min}^{(i)})_j$ with the opposite velocity.

Evaluating $\chi^2(\Theta)$ for a given set of fit parameters is computationally expensive, as it involves solving an S-matrix bootstrap problem. To efficiently estimate the minimum of the $\chi^2(\Theta)$ function in fewer steps, we use a modified version of the standard PSO, called adaptive PSO. We use velocity information to adjust the value of $\omega$, as described in [51].

The goal is to adjust the friction parameter $\omega$ using a control function. The value of $\omega$ determines how effec-

tively the swarm explores the parameter space. If $\omega$ is too large (of order one), the particles will explore erratically, risking failure to converge on the minimum. If $\omega$ is too small, the particles will freeze early, preventing a broad exploration.

In [51], the authors suggest to look at the total velocity of the entire swarm at each step $n$ defined as

$$\mathbb{V}_n = \frac{1}{n_{\mathrm{p}}|\Theta|} \sum_{i=1}^{n_{\mathrm{p}}} |v_n^{(i)}|, \tag{B1}$$

and compare it with a proposed ideal velocity profile

$$\mathbb{V}_n^{\mathrm{ideal}} = \frac{|\epsilon|}{2}\left(1 + \cos\left(\frac{\pi n}{0.95 N_{\mathrm{iter}}}\right)\right). \tag{B2}$$

If $\mathbb{V}_{n-1} \geq \mathbb{V}_n^{\mathrm{ideal}}$, we set $\omega_n = \max(\omega_{n-1} - \Delta\omega, \omega_{\min})$, otherwise $\omega_n = \min(\omega_{n-1} + \Delta\omega, \omega_{\max})$. In our case, $\omega_{\max} = 0.8$, $\omega_{\min} = 0.3$, and $\Delta\omega = 0.1$. We also set $c_1 = c_2 = 1.5$.

The expression of $\mathbb{V}^{\mathrm{ideal}}$ contains the parameter $N_{\mathrm{iter}}$ which is the max number of iterations of the search. The ideal velocity is a monotonic decreasing function that starts at $|\epsilon|$ when $n = 0$. As we approach the end of the run when $n \sim N_{\mathrm{iter}}$, the ideal velocity is so small that it forces the friction to decrease to its minimum value $\omega_{\min}$

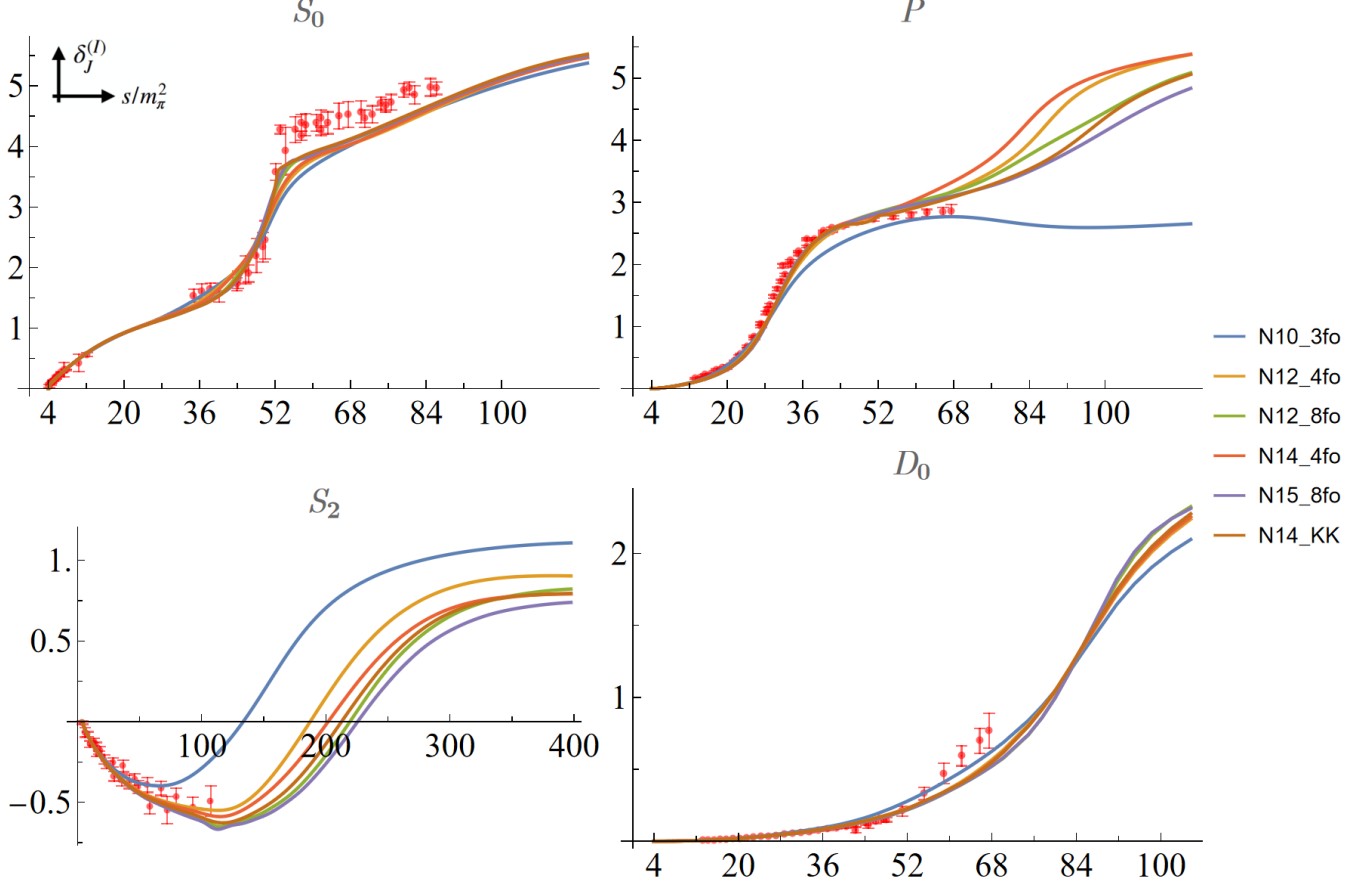

FIG. 15: The phase shift $\delta_0^{(0)}$, $\delta_1^{(1)}$, $\delta_0^{(2)}$ for 6 computations with different foliation parameter. See Table IV for the detailed parameters of these computations, as well as the complex mass of corresponding the isospin 2 tetraquark meson. Red: Experiment data.

hence freezing the search in a neighbor of the global best found up to that point.

The risk with this approach is that choosing a small $N_{\text{iter}}$ could lead to getting stuck in a local minimum. To mitigate this, we performed two searches with different values of $N_{\text{iter}} = 30$ and $60$.

Figure 14 shows the projection of particle positions onto the $(z_0, z_2)$ plane at various steps $n$ during a run with $N_{\text{iter}} = 60$. Black dots represent the particles' current positions $\Theta_n^{(i)}$ at step $n$, blue dots indicate each particle's best position $X_n^{(i)}$ up to that step, and the red dot marks the global best position $Y_n$ found by the swarm. At $n = 0$, the black and blue dots coincide. The gray region represents the search domain.

## Appendix C: More computations for various foliation parameters

To test the numerical robustness of our physical predictions with respect to the choice of basis, we performed other 6 runs with different foliation parameters. We fixed meson masses to the best-fit parameters obtained from the particle swarm optimization. Specifically the masses of $\rho(770)$, $f_0(980)$, $f_0(1370)$, $f_2(1270)$ are set to $\{5.54426 + 0.538798i, 7.14996 + 0.261433i, 9.50987 + 1.81665i, 9.31046 + 0.645446i\}$ respectively. Table IV lists the foliation parameters and number of terms in the ansatz. The resulting phase shifts are shown in Figure 15. We also found the mass of the isospin-2 tetraquark on the complex plane, which is reported in the last column of Table IV.

These results demonstrate that the spectrum and phase-shift predictions quoted in the main text are robust with respect to various foliation parameters. We also observed slight improvement of local behavior and convergence once we include the $K\bar{K}$ singularity.

| Label | $(N, M)$ | Foliation | $L_{max}$ | $N_{var}$ | $I = 2$ tetraquark mass |
|-------|----------|-----------|-----------|-----------|--------------------------|
| N10_3fo | $(10, 8)$ | $\{20/3, 30, 50\}$ | 12 | 211 | $12.6854 + 1.9611\,i$ |
| N12_4fo | $(12, 10)$ | $\{20/3, 30, 50, 86\}$ | 12 | 397 | $14.1138 + 2.28349\,i$ |
| N12_8fo | $(12, 10)$ | $\{20/3, 30, 50, 86, 116, 150, 180, 210\}$ | 12 | 757 | $14.9907 + 2.39145\,i$ |
| N14_4fo | $(14, 12)$ | $\{20/3, 30, 50, 86\}$ | 12 | 547 | $14.4221 + 2.49791\,i$ |
| N15_8fo | $(15, 13)$ | $\{20/3, 30, 50, 86, 116, 150, 180, 210\}$ | 18 | 1214 | $15.0575 + 2.4908\,i$ |
| N14_KK | | $\{(4, 20/3, 14), (4, 30, 12), (4, 50, 12), (4, 86, 12), (52.13, 86, 4)\}$ | 12 | 565 | $14.7698 + 2.42318\,i$ |

TABLE IV: Number of terms and foliation parameters for various computations. In the last row, each triplet $\{b, \sigma, N\}$ denotes the branch point position, foliation center, and number of $\alpha$ and $\beta$ parameters. The last column is the complex mass for the isospin 2 tetraquark meson.

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
