# Peer review of "From data to the analytic S-matrix: A Bootstrap fit of the pion scattering amplitude"

_SciPost Physics_

## Round 1 · Referee Report · Aninda Sinha (Referee 2) · 2025-11-18

Strengths
- Provides a novel, robust technique relying on basic consistency criteria that an S-matrix needs to satisfy to constrain pion scattering, relying on some minimal experimental inputs.
- Thorough examination of low energy predictions from this novel fit program and comparison with experimental results to demonstrate reliability.
- Novel prediction for tetraquark state
- Appendices do a reliable check on numerics
Weaknesses
- Experimental inputs used need to be highlighted better.
- The error bars on the results need clarification.
Report
Requested changes
-
The authors mention on pg 4 "It turns out that to correctly reproduce the experimental data, we will need to consider at least four resonance zero constraints, one for the ρ(770) in the P wave, the two f0’s in the S0, and one f2 in the S2 wave". This seems to be an important point that should be mentioned earlier on, in my opinion. The wording around this sentence is also a potential source of confusion. For instance, the authors mention in the previous sentences "At this point, it is important to stress that we are agnostic about the values of both the chiral and resonance zeros. In our construction, we only assume their existence with the corresponding quantum numbers. Their numerical values will be a prediction of the fit. For the spectrum, we only assume the existence of a part of it" This makes it a bit unclear if all of rho(770), the two f0's and the f2 have to be put in from experiments by hand. [The wording in the previous paragraph suggests no but this is not completely clear].
-
I am confused by the error bars. Fig 15, P and S2 seem to give a spreading of values depending on the foliation parameters. It does not seem that tables 1,2,3 have taken this spread into account. If not this needs to be clarified.
Recommendation
Publish (easily meets expectations and criteria for this Journal; among top 50%)
Strengths
1- Tetraquark new prediction. 2- New AI method (PSO) in the numerical toolbox 3- Successful input of experimental multi-particle data 4- Match to chiPT a posteriori 5- Improved understanding of the sigma resonance
Report
The paper easily meets criteria for publication in scipost physics.
Requested changes
One thing that was maybe not discussed: how sensitive is the amplitude and the tetraquark prediction in particular to the choice of inelastic profile? A comment on this would be helpful, as an analysis of this fact would probably require an entire separate study.
It wasn't immediately clear to me how you could do any assumption on the spectrum? How exactly are the resonance input? As exact zeros of the partial waves in the complex plane? That sounds like a dangerous numerical procedure?
Fig. 3 is not so easy to understand on its own, for instance why would a big red cross appear to lie outside of "allowed space", or any kink for that matter. I understand the matter is complex, but maybe the figure or the caption could be improved.
Could you comment more precisely the number of parameters you're fixing and their effect? You mention that it would be good to let the bootstrap figure them all out, but why exactly did you chose the specific parameters you fixed?
Recommendation
Publish (surpasses expectations and criteria for this Journal; among top 10%)
Author: Kelian Häring on 2025-11-25 [id 6072]
(in reply to Report 1 on 2025-11-07)The comment author discloses that the following generative AI tools have been used in the preparation of this comment:
We used ChatGPT 5.1 to correct grammar and perform spellcheck on our answer.
We thank the referee for the positive assessment of our work and for the helpful comments. We have addressed the requested clarifications and made minor revisions accordingly. A detailed response is provided below.
Question 1
One thing that was maybe not discussed: how sensitive is the amplitude and the tetraquark prediction in particular to the choice of inelastic profile? A comment on this would be helpful, as an analysis of this fact would probably require an entire separate study.
We thank the referee for this valuable question. As briefly mentioned in the manuscript (last paragraph of p. 9), while we expect that changing the inelastic profile may slightly shift the position of the tetraquark, we expect that it would not challenge its existence or overall prediction. The physical intuition comes from the interpretation of the imaginary part of the zero as the width of the particle which would be affected by inelasticity while its mass remains almost unchanged.
To provide a first estimate of this sensitivity, we included in Fig. 10 the impact on the signal obtained by extrapolating the inelasticity in the $I=2$, $\ell=0$ channel. A full exploration of the dependence on different inelastic profiles would indeed require a more extensive analysis, which we believe is beyond the scope of the present study but would be an interesting direction for future work.
Question 2
It wasn't immediately clear to me how you could do any assumption on the spectrum? How exactly are the resonance input? As exact zeros of the partial waves in the complex plane? That sounds like a dangerous numerical procedure?
We thank the referee for this question and are happy to clarify the procedure. In each bootstrap optimization, we assume the presence of specific resonances in the amplitude, two in the $S_0$-wave and one each in the $P$- and $S_2$-waves. These resonances are implemented by imposing a zero of the corresponding $S$-matrix in the complex plane of each channel.
Once this condition is enforced, the particle swarm optimization is used to determine the optimal position of these zeros so as to best fit the data. In this way, while the existence of resonances is assumed, their locations are not fixed a priori but are instead dynamically determined during the fitting procedure.
We acknowledge that imposing zeros in the complex plane may appear numerically delicate; however, in our implementation it remains stable and reliable within the range of parameters explored.
Question 3
Fig. 3 is not so easy to understand on its own, for instance why would a big red cross appear to lie outside of "allowed space", or any kink for that matter. I understand the matter is complex, but maybe the figure or the caption could be improved.
We thank the referee for pointing this out. We agree that the figure was not sufficiently self-contained. Following the referee’s suggestion, we have improved the color scheme and clarified the caption to better explain the role of each element in the plot.
Question 4
Could you comment more precisely the number of parameters you're fixing and their effect? You mention that it would be good to let the bootstrap figure them all out, but why exactly did you chose the specific parameters you fixed?
We thank the referee for this question and agree that our choice of parameters is, a priori, not obvious. Since the space of observables is in principle infinite, selecting an appropriate set of fitting parameters requires some physical motivation. Below we clarify the parameters used and their role in the fitting procedure.
- Two chiral zeros: $(z_0, z_2)$. Their are included to enforce $\chi$PT.
- Three low-energy constants: $(a_{0}^{(0)},a_{0}^{(2)}, a_{1}^{(1)})$. These correspond to the leading low-energy constants of the amplitude and are commonly used in the pion-scattering literature. They are employed to scan the space of possible amplitudes and capture the dominant threshold behavior.
- Four complex resonances masses: $(m_\rho^2,m_{f_0}^2,m_{f_0'}^2, m_{f_2}^2 )$. They capture the peaks in the amplitude and the jumps in the phase shifts. For instance, if these resonances are removed, scanning over the remaining five parameters allows us to reproduce the low-energy phase in the $S_0$ channel, but fails to capture the jump around $s \approx 50$.
In future work, we intend to relax the assumptions on the resonance spectrum. One way to achieve this is to enlarge the space of low-energy parameters considered within the bootstrap. We plan to explore this direction in subsequent studies.

Author: Kelian Häring on 2025-11-25 [id 6073]
(in reply to Report 2 by Aninda Sinha on 2025-11-18)The comment author discloses that the following generative AI tools have been used in the preparation of this comment:
We used ChatGPT 5.1 to check the grammar and spelling of our answers to the referee.
We thank the referee for the encouraging feedback. Minor revisions have been made following their suggestions. Our responses are given below.
Question 1
We thank the referee for pointing out the potential confusion in the wording regarding which resonances are assumed and which are dynamically generated. We agree that this clarification is important. We have therefore revised the paragraph to make it explicit.
Question 2
The spread observed in Fig. 15 originates primarily from the smallest ansatz size, with $N = 10$ and $M = 8$. In the main text, however, we relied exclusively on the largest ansätze. As illustrated in Fig. 15, once $N \geq 12$ the dependence on the foliation parameters becomes significantly smaller, and its impact on the final results is minimal and taken into account in the errors quoted in the main text (e.g. in table 1 and 2).
We also remind the referee that in the main text we explicitly highlighted results that remain subject to these systematic uncertainties, for instance, using an asterisk in Fig. 1 to indicate values sensitive to the foliation choice.

---

## Round 2 · Author Response

Dear Editor,
We are pleased to resubmit our manuscript "From data to the analytic S-matrix: A Bootstrap fit of the pion scattering amplitude" for publication in SciPost Physics.
We would like to thank the editor and the referees for their positive feedback and the constructive comments made on our work. Following their questions and comments we modified the manuscript as detailed in "List of changes". The referee’s questions have been addressed in the dedicated section.

We believe that the revised version addresses the question raised and improves the clarity of the manuscript.

Kind regards,
Kelian Häring (on behalf of the authors)

---

## Round 2 · List of Changes

• Figure 3: We have improved the color scheme and clarified the caption to better explain the role of each element in the plot.
  • Section II.C: We rephrased section II.C starting from "The physical spectrum is...". We agree that the previous version was potentially confusing regarding which resonances are assumed and which are dynamically generated. We have therefore revised the paragraph to make this distinction explicit.

---

## Editorial Decision

accepted_in_target_journal